# A true choice of place of birth? Swiss women's access to birth hospitals and birth centers

Sebastian Rauch[1]*, Louisa Arnold[2], Zelda Stuerner[3], Juergen Rauh[1], Michael Rost[3]*

**1** Institute of Geography and Geology, University of Wuerzburg, Wuerzburg, Germany, **2** Institute of Psychology, Friedrich-Schiller-University of Jena, Jena, Germany, **3** Institute for Biomedical Ethics, University of Basel, Basel, Switzerland

* michael.rost@unibas.ch (MR); sebastian.rauch@uni-wuerzburg.de (SR)

**Data Availability Statement:** The present study employed data from various sources. While highly disaggregated population data were used under license (University of Basel) for the current study and are not publicly available, all other data were

## Abstract

While the place of birth plays a crucial role for women's birth experiences, the interest in out-of-hospital births has increased during the Covid-19 pandemic. Related to this, various international policies recommend enabling women to choose where to give birth. We aimed to analyze Swiss women's choice between birth hospitals and birth centers. Employing spatial accessibility analysis, we incorporated four data types: highly disaggregated population data, administrative data, street network data, addresses of birth hospitals and birth centers. 99.8% of Swiss women of childbearing age were included in the analysis (N = 1.896.669). We modelled car travel times from a woman's residence to the nearest birth hospital and birth center. If both birth settings were available within 30 minutes, a woman was considered to have a true choice. Only 58.2% of women had a true choice. This proportion varied considerably across Swiss federal states. The main barrier to a true choice was limited accessibility of birth centers. Median travel time to birth hospitals was 9.8 ($M = 12.5$), to birth centers 23.9 minutes ($M = 28.5$). Swiss women are insufficiently empowered to exercise their reproductive autonomy as their choice of place of birth is significantly limited by geographical constraints. It is an ethical and medical imperative to provide women with a true choice. We provide high-resolution insights into the accessibility of birth settings and strong arguments to (re-)examine the need for further birth centers (and birth hospitals) in specific geographical areas. Policy-makers are obligated to improve the accessibility of birth centers to advance women's autonomy and enhance maternal health outcomes after childbirth. The Covid-19 pandemic offers an opportunity to shift policy.

## Introduction

Over the past decades, there has been a–sometimes keenly thought–debate about planned place of birth which mostly has been centered around infant and maternal safety, yielding spectrum-spanning opinions on the issue [1–4]. While some argue that planned home birth "has unnecessary, preventable, irremediable increased risk of harm for pregnant, fetal, and neonatal patients" [3, p31] and that "the obstetrician should recommend planned hospital birth" [5], others raise the question whether we, perhaps, need to ask "is hospital birth safe or

open source. Population data are only available from the Swiss Federal Statistical Office. Thus, the dataset generated and analyzed during the current study can neither be made publicly available nor shared with others, since it employed the above mentioned population data. That means there are legal restrictions to sharing our data both publicly and privately. However, further results for smaller administrative areas (e.g. communities, cities) or comparisons between different geographical areas can be made available by the corresponding author upon reasonable request. Moreover, interested readers can contact the Swiss Federal Statistical Office and order the respective populations statistics (https://www.bfs.admin.ch/bfs/en/home/statistics/population/surveys/statpop.html) which then can be combined with the other three data sources that are open source (see methods section), thereby creating the data set generated and analyzed during the current study.

**Funding:** This publication was supported by the Open Access Publication Fund of the University of Wuerzburg. This work was also supported by the Research Fund Junior Researchers, University of Basel (https://www.unibas.ch/en/Research/Financing/Research-Fund.html). The funding source had no involvement in the conduct of the research.

**Competing interests:** The authors have declared that no competing interests exist.

sustainable for low risk women?" and advocate for a broadening of our understanding of safety "including physical, psychological, social, cultural and spiritual safety" [1, p9/10]. Besides, it appears that various biases (e.g. publication bias, professional bias) hamper production and use of evidence surrounding the risks associated with place of birth for well-informed clinical decision-making [4]. Irrespective of the debate around place of birth, it has to be noted that in response to the Covid-19 pandemic many families have reassessed their birth plans and awareness of and interest in out-of-hospital births (e.g. home birth, birth centers) have increased, catalyzing the use of home-birthing and birth centers [6–9].

Recent empirical evidence from systematic reviews and meta-analyses has demonstrated that risk of perinatal or neonatal mortality was not different when birth was intended at home or in hospital and that low-risk women planning a home birth experienced less interventions and less adverse maternal outcomes [10–12]. Lastly, results of a prospective British birthplace study have lent further support to the claim that healthy women with low-risk pregnancies should be offered a choice of birth setting [13]. In light of this evidence, today various policies recommend enabling women to choose where to give birth and improving women's access to birth centres [14–17]. For example, according to a recent British guideline on intrapartum care, it should be (a) ensured that all birth settings are available to all women in the local or in a neighboring area; (b) explained in an encouraging way to both multiparous and nulliparous women that they can choose any birth setting; (c) advised to low risk nulliparous women that giving birth in a midwife-led unit "is particularly suitable for them because the rate of interventions is lower and the outcome for the baby is no different compared with an obstetric unit" [18, p.1]. These claims were endorsed by the Swiss Association of Midwifes [14].

Although understood differently across various contexts, respect for autonomy is a central principle in medical ethics being instructive for clinical practice and research [19]. Undoubtedly, it mandates being able to make reasoned choices without interference. Being able to exercise reproductive rights and, particularly, to determine how and where to give birth qualifies as reproductive autonomy [20]. Similarly, a concept analysis on autonomy in place of birth emphasized that being capable of choosing place of birth in the absence of coercion is one prerequisite for an autonomous choice [21]. Finally, emphasizing the imperative to enable women to make reasonable choices, a Cochrane review on place of birth has pointed out that women residing in areas where they are not well informed about possible places of birth "may welcome ethically well-designed trials that would ensure an informed choice" [22, p2]. However, women's choices of place of birth are limited by, amongst others, geographical distance from home to the preferred birth setting, by the availability of midwives for home births, or by a family's low economic status rendering the use of birth centers unaffordable.

The corollary is that policy-makers, obstetric care providers, and governmental authorities are obligated to create a supply level that empowers women to have a true choice, which means (besides being well-informed) having reasonable access to all birth settings. Therefore, the purpose of this paper is to analyze geographical constraints to Swiss women's choice between a birth hospital and a birth center to explore inequalities, to identify underserved areas, and to estimate the proportion of Swiss women having a true choice of place of birth in times of Covid-19. We employed spatial accessibility analysis (SAA) with a focus on car travel time from a woman's residence to the respective birth settings. Our findings provide valuable insights into how reproductive autonomy of women of childbearing age is limited by insufficient accessibility of birth settings at the time of choice of place of birth.

## Methods

### Spatial accessibility

One measure of spatial accessibility to healthcare is distance to nearest provider, which is usually measured from a patient's residence or from the centroid of a spatial raster-cell [23]. Typically, the statistic-based strategies model the spatial separation between places (e.g. birth settings) and people (e.g. women of childbearing age) analyzing distance or time. The underlying aim is to improve accessibility to healthcare.

### Swiss setting

In Switzerland, the most common place of birth is the hospital (2017: 98.3% of 85.990 births) [24]. Besides hospitals, women can give birth at home or in birth centers: freestanding, midwife-led, primary care facilities in the public health system which might collaborate with hospitals and private gynecologists (e.g. regarding transfer, consultation), but in which no physicians work. As such, care is entirely performed by midwifes. Their work is based on guidelines for birth centers.

### Defining (true) choice of place of birth

The journey time threshold for birth centers and birth hospitals was set at 30 minutes as this represents an acceptable travel time and this cut-off was used in similar research to define reasonable access to birth facilities [16]. For the purpose of this paper, if both the nearest birth hospital and the nearest birth center were accessible within 30 minutes, a woman was considered to have a true choice.

### Data

We incorporated data from four different sources: (1) highly disaggregated population data (i.e. 100x100 meters raster-cells) from the Swiss Federal Statistical Office containing information on gender and age group composition of the resident population to spatialize the researched population [25]; (2) administrative data from the Swiss Federal Office of Topography Swisstopo to spatialize communities and federal states [26]; (3) street network data from OpenStreetMap to which traffic-related car travel speeds were assigned; (4) addresses of existing birth hospitals from the Swiss Federal Office of Public Health and of birth centers from the Interest Group of Swiss Birth Centers (as of December 2020) [27, 28]. Apart from (1) which could be utilized through a University license, the remaining databases were open source and did not require any permission. The suitability of these data has been demonstrated in comparable research [29–31].

### Spatial accessibility analysis

Using ArcGIS (ESRI, Redlands, CA), we analyzed Swiss women's access to birth centers and birth hospitals with the variable of interest being car travel time from women's residence to the respective birth setting for each 100x100 meters population raster-cell in Switzerland (N = 254.278). The analysis was limited to women of childbearing age which the World Health Organization defines as aged 15–49 [32]. Our analysis included 99.8% of Swiss women of childbearing age (N = 1.896.669). For data protection reasons (i.e. rendering the identification of a single person impossible), each raster-cell of the available population data with an actual value in range of"1" to "3" (i.e. less than four persons residing) got assigned the value "3", which leads to an artificial increase of the overall N. Thus, data preparation required correction of counts for the respective raster-cells.

In SAA, besides the given infrastructure, the estimated value for travel speed is the second key factor. Hence, in addition to the 27 street categories used in OpenStreetMap, we further differentiated streets based on the surrounding population density and on speed limit. Subsequently, streetmap data was transferred into a routing network. The latter was then combined with the disaggregated 100x100 meters population data (i.e. age and gender composition), which allowed analyzing population on a fine spatial resolution. This combination of the routing network and the population data provided evidence on the supply level of birth hospitals and birth centers. In total, 108 birth hospitals and 25 birth centers were included. We employed two approaches to model accessibility, both resting on the nearest center hypothesis (an individual chooses the facility that is closest to its residence) [29, 30].

First, we used a raster-based method [30]. Based on the streetmap data, we calculated the time needed from each raster-cell's centroid to the 133 facilities, resulting in an overall accessibility matrix. Travel time of each centroid of a raster-cell was assigned to all women of childbearing age residing in the respective raster-cell. Second, we used a vector-based method [30]. Isochronic values were set to determine in which areas women could reach a facility within the acceptable time threshold of 30 minutes [16]. The resulting isochronic areas indicate a reasonable access for women and enabled us to identify areas with (in)sufficient access (Maps 2 and 3). Most importantly, this step revealed areas with a true choice between birth hospitals and birth centers by detecting overlapping isochronic areas, that is both birth settings could be reached within 30 minutes (Map 1). In the corresponding map, we not only illustrated the number of women of childbearing age but weighted them by the relative proportion of births of each age group in the total number of annual births in Switzerland (15–19: 3590.430%; 20–24: 5.2186.260%; 25–29: 20.20624.243%; 30–34: 33.43340.113%; 35–39: 22.35626.823%; 40–44: 5.3506.419%; 45–49: 4250.510%) [25], thereby providing a visual representation of estimated births.

## Statistical analysis and visualization

We exported quantitative data of SAA from ArcGIS to SPSS.26 (SPSS Inc., Chicago, IL) and employed descriptive statistical analyses in order to obtain a better understanding of accessibility of birth settings. Results were visualized to better determine spatial and temporal conditions of access to birth settings.

## Results

At a national level, 58.2% of women had a true choice between birth hospital and birth center. Overall, birth hospitals' accessibility was better. Median travel time to birth hospitals was 9.8 minutes, to birth centers 23.9 minutes. Mean travel time to birth hospitals was 12.5 minutes, to birth centers 28.5 minutes (Fig 1). The proportion of women with a true choice, quartiles of travel times to both birth settings, minimum and maximum travel times, and difference between mean travel times are presented in Table 1.

### True choice of birth setting

While at a national level less than six out of ten women of childbearing age had a true choice, this proportion varied considerably across cantons (Swiss states; Table 1), ranging from four cantons with 90–100% of women with a true choice (Nidwald, Basle-City, Basle-Country, Geneva) to five cantons with less than 5% of women (Uri, Schwyz, Glarus, Zug, Grisons). Map 1 depicts geographical areas in which women had a true choice.

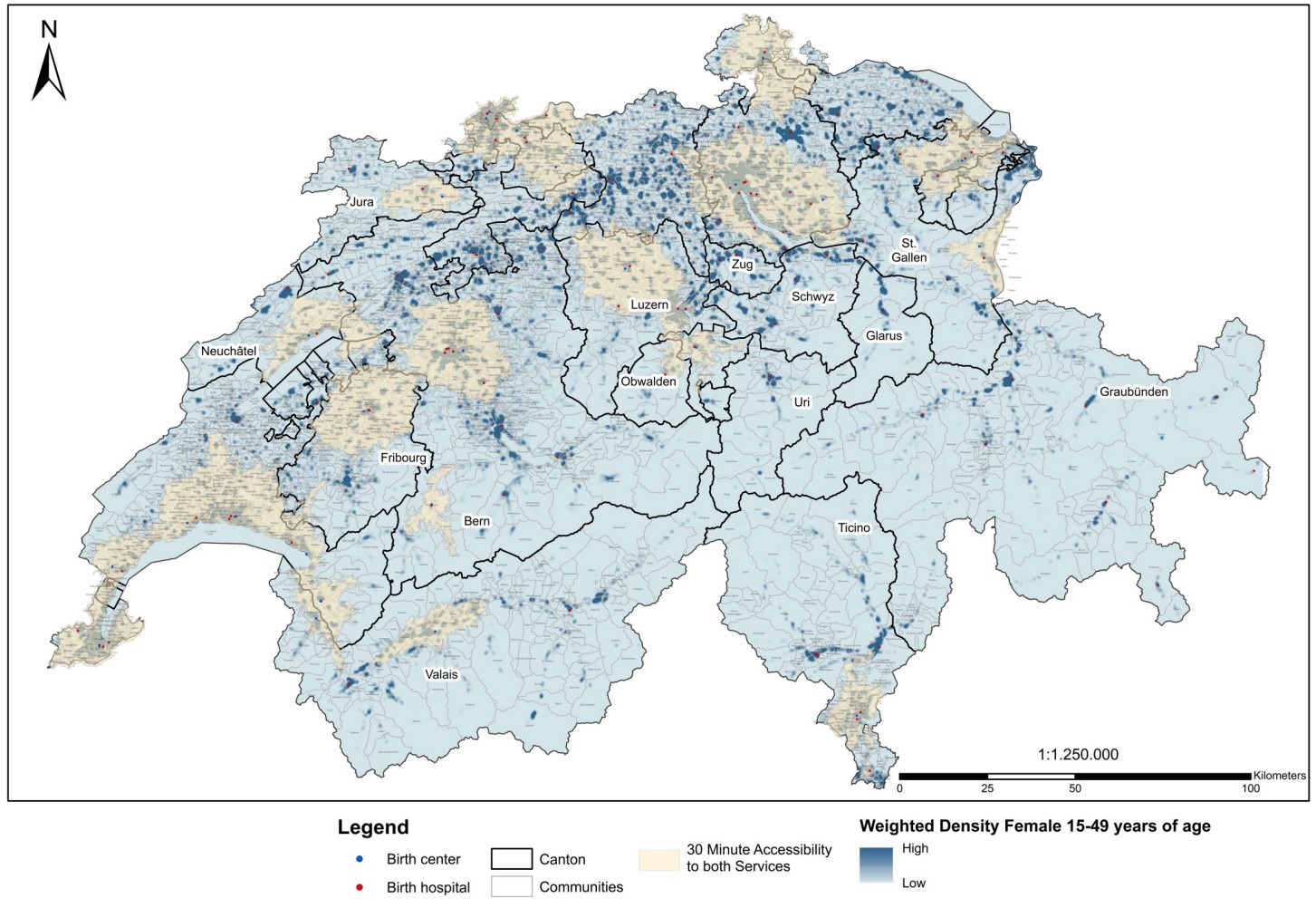

**Map 1. Geographical areas with true choice of birth setting.**

### Accessibility of birth hospitals and birth centers

For Switzerland, interquartile range of accessibility of birth hospitals was 11.7 minutes, that is 50% of women of childbearing age had a travel time to the nearest birth hospital between 5.2 and 16.9 minutes (Table 1). Unsurprisingly, the supply level of birth hospitals was very good with 94% of women having reasonable access to a birth hospital within 30.1 minutes. Consequently, variation across cantons was comparatively small. Map 2 depicts accessibility of birth hospitals on a continuum from red to green; yellow indicates the 30 minutes threshold.

For Switzerland, interquartile range of accessibility of birth centers was 28.9 minutes, that is 50% of women of childbearing age had a travel time to the nearest birth center between 10.9 and 39.8 minutes. The supply level of birth centers was worse than the one of birth hospitals with only 59% of women having a reasonable access to a birth center within 30.2 minutes. Variation across cantons was enormous, ranging from seven cantons in which at least 75% of women of childbearing age had access to a birth hospital in less than 30 minutes (Zurich, Nidwald, Basle-City, Basle-Country, Schaffhausen, Vaud, Geneva) to nine cantons in which at least 75% of women of childbearing age only had access to a birth center in more than 30

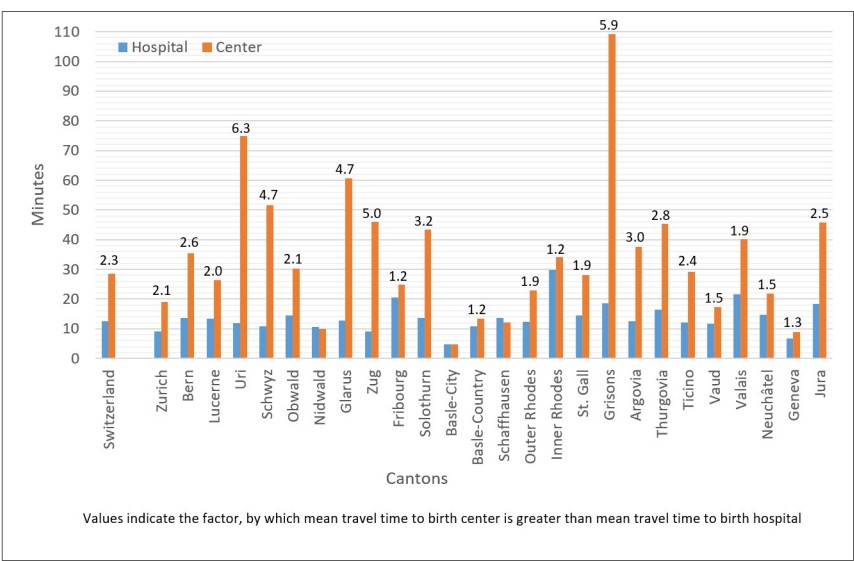

**Fig 1. Mean travel times in minutes for both birth settings.**

minutes (Uri, Schwyz, Glarus, Zug, Solothurn, Inner Rhodes, Grisons, Argovia, Thurgovia). For four cantons, 0% of women had access within less than 30 minutes (Uri, Glarus, Zug, Grisons). Map 3 depicts accessibility of birth centers.

## Difference between accessibilities of birth settings

For Switzerland, difference between mean travel times of birth settings was 16.0 minutes, with birth centers' accessibility being worse. Again, this difference varied considerably across cantons, ranging from ten cantons with a difference smaller than 10 minutes (Zurich, Nidwald, Fribourg, Basle-City, Basle-Country, Schaffhausen, Inner Rhodes, Vaud, Neuchâtel, Geneva) to five cantons with a difference bigger than 30 minutes (Uri, Schwyz, Glarus, Zug, Grisons); for women in Grisons (N = 40.716), mean travel time to the next birth center was more than 1.5 hours longer. Map 4 depicts the difference between travel times (birth center–birth hospital); green indicates a difference between -10 and 10 minutes, that is an area where both accessibilities can be considered marginally different; blue indicates better accessibility of birth centers; red of birth hospitals. Fig 1 presents mean differences for Switzerland and cantons and the factor by which travel times to birth hospitals are greater than travel times to birth centers.

## Discussion

Overall, women of childbearing age in Switzerland are insufficiently empowered to exercise their reproductive autonomy as their choice of place of birth is significantly limited by geographical constraints. SAA revealed that 41.8% (N = 792.807) of women lack reasonable access to both birth settings and therefore do not qualify as having a true choice. This is a grievance for numerous reasons.

First, providing women with a true choice is an ethical imperative that derives from women's reproductive rights and the corresponding respect for their reproductive autonomy [20, 33]. Second, according to medical guidelines, women should be enabled to have the full range of choice since there are well-described benefits of giving birth in birth centers (e.g. for low-

**Table 1. Accessibility of birth settings at national and cantonal level.**

| | | | Travel times in min.[1] | | | | | | | | | | |
| | | | To birth hospitals | | | | | To birth centers | | | | | |
| | N (women 15–49) | True choice | Min | Q1 | Mdn | Q3 | Max | Min | Q1 | Mdn | Q3 | Max | Mean Δ[2] |
|---|---|---|---|---|---|---|---|---|---|---|---|---|---|
| **Switzerland** | **1.896.669** | **58.2%** | **0.0** | **5.2** | **9.8** | **16.9** | **148.1** | **0.0** | **10.9** | **23.9** | **39.8** | **335.3** | **16.0** |
| Zurich | 354.560 | 80.0% | 0.0 | 5.2 | 8.5 | 12.2 | 45.9 | 0.0 | 10.7 | 17.7 | 27.5 | 51.2 | 9.9 |
| Bern | 219.957 | 40.7% | 0.0 | 5.0 | 10.7 | 19.1 | 88.6 | 0.1 | 16.0 | 35.9 | 48.7 | 133.2 | 21.8 |
| Lucerne | 91.869 | 70.1% | 0.0 | 7.3 | 11.4 | 17.5 | 91.1 | 0.3 | 19.7 | 25.1 | 31.5 | 123.5 | 12.9 |
| Uri | 7.475 | 0.0% | 0.0 | 4.2 | 6.6 | 14.5 | 96.4 | 32.3 | 68.3 | 71.2 | 75.9 | 152.5 | 62.9 |
| Schwyz | 33.699 | 3.6% | 0.0 | 4.5 | 9.0 | 17.6 | 55.3 | 24.6 | 36.8 | 45.6 | 64.8 | 131.0 | 40.8 |
| Obwald | 7.923 | 54.0% | 0.0 | 6.9 | 11.3 | 20.0 | 59.1 | 10.8 | 20.9 | 27.7 | 38.0 | 80.0 | 15.7 |
| Nidwald | 8.654 | 99.1% | 0.0 | 6.0 | 10.9 | 13.6 | 40.3 | 0.3 | 5.7 | 10.2 | 12.5 | 39.0 | -0.8 |
| Glarus | 8.279 | 0.0% | 0.4 | 4.9 | 12.8 | 16.5 | 44.8 | 42.0 | 50.5 | 58.1 | 64.4 | 105.6 | 47.9 |
| Zug | 27.830 | 0.0% | 0.0 | 3.8 | 6.4 | 11.3 | 32.4 | 31.5 | 42.6 | 44.2 | 46.8 | 71.1 | 36.7 |
| Fribourg | 73.567 | 60.8% | 0.0 | 11.5 | 19.7 | 28.7 | 73.1 | 0.1 | 12.7 | 25.1 | 35.8 | 79.1 | 4.3 |
| Solothurn | 57.417 | 12.8% | 0.0 | 6.4 | 11.1 | 17.8 | 64.2 | 14.3 | 33.3 | 44.7 | 54.0 | 74.6 | 29.8 |
| Basle-City | 44.980 | 99.9% | 0.0 | 3.2 | 4.1 | 5.1 | 18.7 | 0.2 | 2.9 | 4.1 | 5.5 | 19.6 | 0.0 |
| Basle-Country | 58.654 | 90.8% | 0.0 | 5.2 | 7.9 | 14.4 | 51.8 | 0.0 | 6.7 | 11.4 | 16.6 | 58.2 | 2.5 |
| Schaffhausen | 16.783 | 83.7% | 0.0 | 5.9 | 8.7 | 19.9 | 45.1 | 0.0 | 3.9 | 6.3 | 20.3 | 42.0 | -1.5 |
| Outer Rhodes | 11.063 | 72.3% | 0.0 | 4.6 | 12.1 | 17.0 | 49.6 | 4.9 | 15.8 | 20.2 | 30.3 | 61.2 | 10.7 |
| Inner Rhodes | 3.343 | 10.2% | 0.0 | 29.2 | 31.6 | 34.4 | 50.7 | 13.8 | 32.5 | 34.1 | 37.0 | 53.9 | 4.3 |
| St. Gall | 110.936 | 49.9% | 0.0 | 6.3 | 12.8 | 20.9 | 53.3 | 0.0 | 14.7 | 29.9 | 41.0 | 79.0 | 13.6 |
| Grisons | 40.716 | 0.0% | 0.0 | 5.4 | 14.4 | 24.1 | 104.8 | 39.8 | 58.7 | 88.5 | 132.4 | 335.3 | 90.7 |
| Argovia | 147.649 | 23.1% | 0.0 | 7.3 | 12.0 | 17.1 | 39.5 | 5.7 | 30.9 | 38.7 | 44.8 | 74.1 | 25.0 |
| Thurgovia | 59.223 | 11.7% | 0.0 | 10.0 | 16.9 | 22.8 | 41.1 | 8.9 | 37.8 | 46.4 | 54.6 | 73.7 | 29.0 |
| Ticino | 72.367 | 50.3% | 0.0 | 4.2 | 8.4 | 13.7 | 148.1 | 0.1 | 9.4 | 29.9 | 39.3 | 188.1 | 17.1 |
| Vaud | 189.100 | 80.5% | 0.0 | 3.9 | 8.2 | 16.2 | 124.3 | 0.0 | 6.9 | 12.4 | 24.4 | 136.3 | 5.7 |
| Valais | 74.291 | 54.8% | 0.0 | 11.4 | 18.4 | 26.2 | 100.2 | 0.0 | 13.1 | 26.5 | 56.6 | 175.6 | 18.6 |
| Neuchâtel | 39.539 | 62.7% | 0.1 | 5.3 | 11.0 | 18.0 | 86.4 | 0.2 | 7.5 | 15.8 | 35.7 | 98.8 | 7.2 |
| Geneva | 121.504 | 98.5% | 0.0 | 3.2 | 6.1 | 8.3 | 28.2 | 0.4 | 5.1 | 7.7 | 11.2 | 37.7 | 2.0 |
| Jura | 15.291 | 47.3% | 0.0 | 6.5 | 13.7 | 28.0 | 113.1 | 0.2 | 13.2 | 40.0 | 77.2 | 149.8 | 27.4 |
| | Map 1 | | Map 2 | | | | | Map 3 | | | | | Map 4 |

*Note.* Values were rounded to first decimal point.

[1] from a woman's 100m x 100m raster-cell's center to the respective birth facility

[2] Δ stands for difference, namely: (mean travel time to birth centers)–(mean travel time to birth hospitals).

risk women the level of interventions is lower for out-of-hospital births without additional risks for the infant, positive association between birth interventions and long-term childhood illnesses) [18, 34–37]. Third, not being able to choose the preferred birth setting increases the likelihood of a mismatch between maternal preferences for and the actual facility culture, which translates itself into a higher risk for conflicts and disagreements during IP-DM (e.g. regarding interventions, birthing position, examinations). It is known that such conflicts and disagreements are one etiological factor for autonomy-depriving IP-DM and mistreatment of women in childbirth [38, 39]. The latter, in turn, were shown to negatively affect psychological maternal outcomes after childbirth (i.e. improved psychological outcomes when maternal autonomy was respected and IP-DM was adequately shared among the involved parties) [40, 41]. Hence, adhering to the bioethical principle of respect for (reproductive) autonomy has a salutogenetic effect during and after birth. Fourth, empirical evidence indicates that

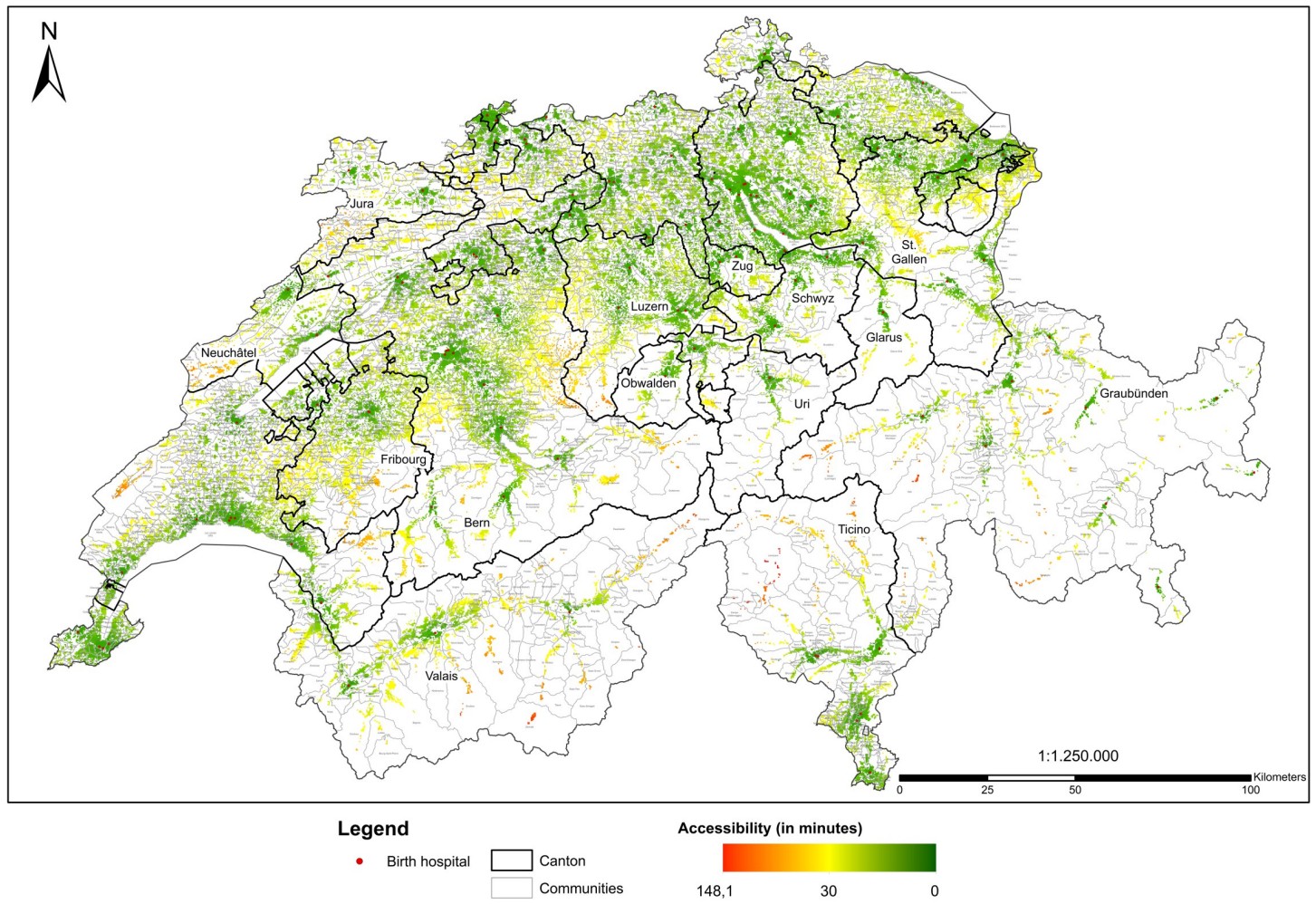

**Map 2. Accessibility of birth hospitals.**

geographical accessibility of place of birth influences women's choices [42], which means that a proportion of women end up delivering at a place primarily because it was close, but less because it was preferred. In fact, a recent systematic review of women's preferences in relation to place of birth concluded that policy-makers and service providers shall extend the availability existing services to offer women a choice that enables them to access services which correspond to their needs and preferences [43]. Along similar lines, a recent resurgence in the use of birth centers in the US was attributed to a "mismatch between what the dominant models of care offer and what most women want" [37, p14]. Finally, given the high numbers of parents experiencing their child's birth, the previous considerations are not only significant at an individual level but also at a public health level.

Over the past years more and more women opened up about instances of mistreatment in childbirth, such as physical or verbal abuse, discrimination, unconsented interventions, loss of autonomy, or informal coercion. The current state of research suggests that mistreatment of women in childbirth occurs regularly and is not limited globally to particular geographical areas [38, 44–47]. Spanning 34 countries, a systematic review from the World Health Organization highlighted that mistreatment can occur at the levels of interpersonal interaction and

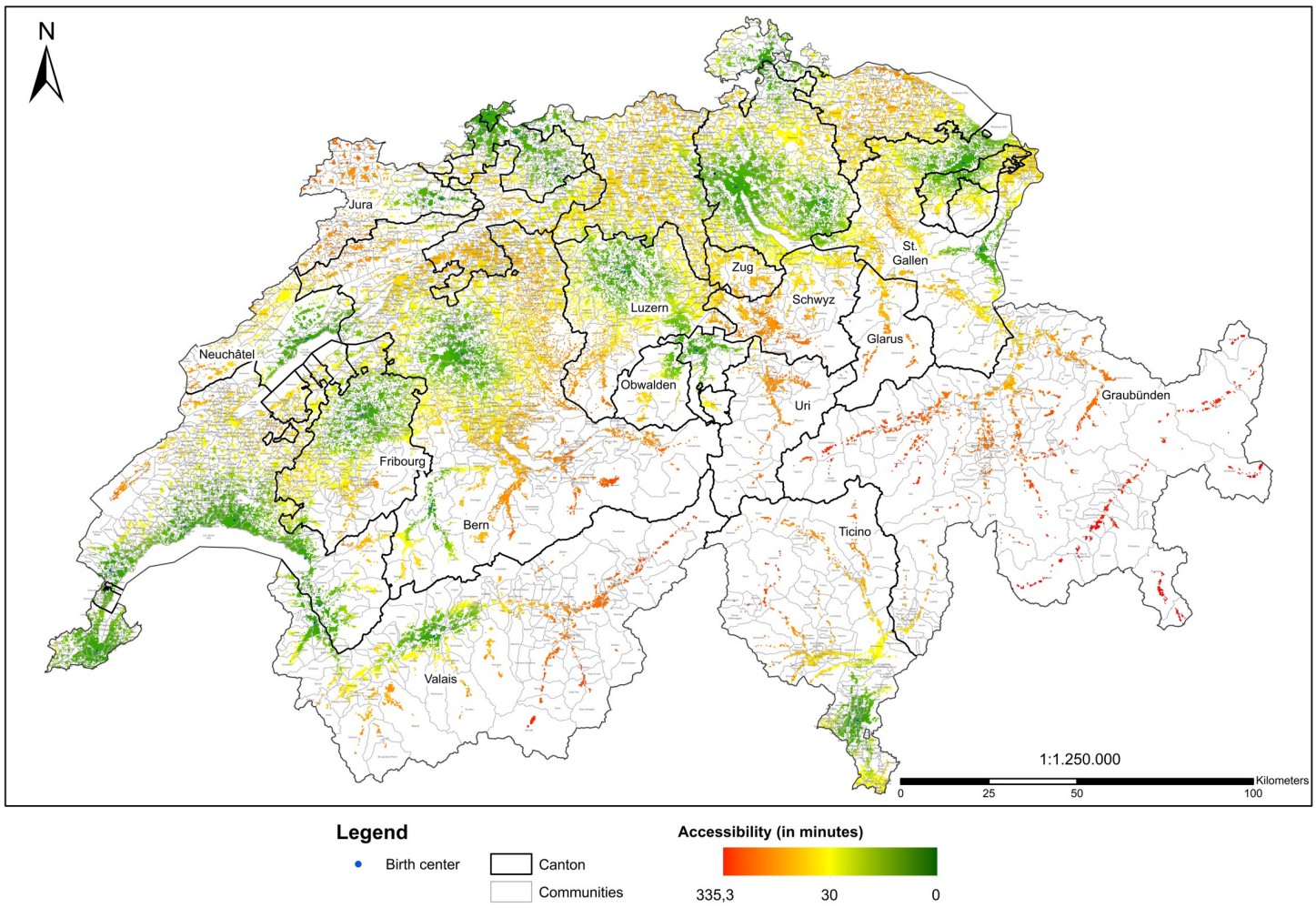

**Map 3. Accessibility of birth centers.**

the facility [44]. In particular, poor rapport between providers and women and a facility's culture were important determinants of mistreatment. Related to this, a recent nationwide Swiss study revealed that 26.7% of women experienced informal coercion in childbirth (i.e. opposition, intimidation, manipulation) and that the likelihood of experiencing informal coercion was higher (a) for in-hospitals births than for births in birth centers, (b) for women with a higher preference for autonomous decisions, and (c) for births with interventions [46]. Moreover, findings from the US found higher rates of mistreatment in hospital settings than in community birth settings such as birth centers and when women's and providers' preferences for care did not match one another [38]. Finally, it has been evidenced that women equally value safety and psychosocial wellbeing during childbirth [48]. Consequently, the place of birth, the facility culture, and the level of alignment of women's and providers' preferences play a crucial role for the quality of women's birth experiences as well as for physical and psychological health outcomes. This mandates to enable women to choose their preferred birth setting.

The main barrier to a true choice of place of birth was the highly limited accessibility of birth centers in some geographical areas. For Switzerland, 20% of women of childbearing age

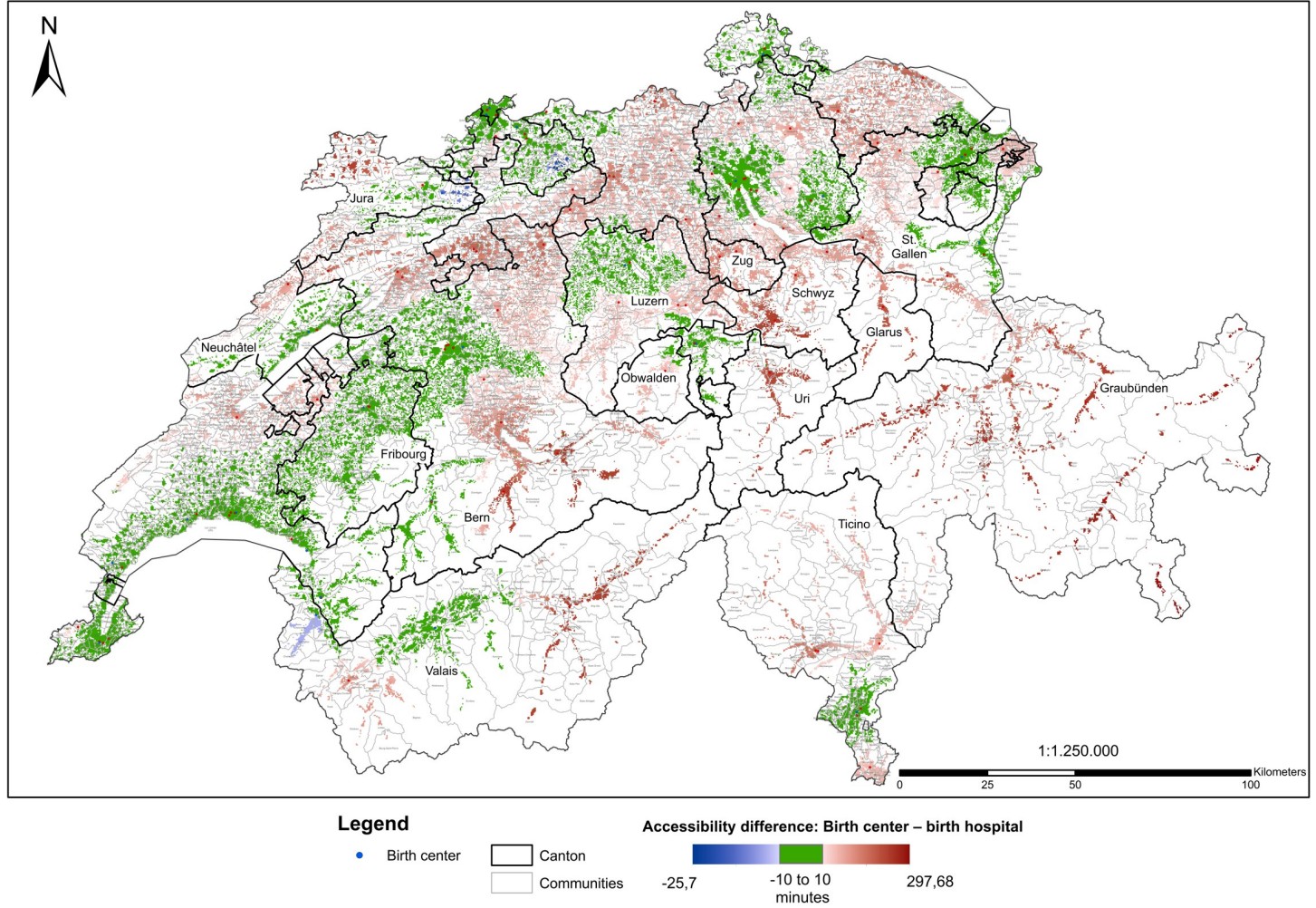

**Map 4. Difference between travel times.**

cannot access a birth center in less than 43.6 minutes and 10% cannot in less than 53.0 minutes. For four cantons the minimum travel time to the nearest birth center was more than 30 minutes (Uri, Glarus, Zug, Grisons) which means that not a single woman of childbearing age had a true choice in these cantons. Furthermore, 50% of women of childbearing age living in Grisons faced a travel time of almost 1.5 hours; in Uri of more than 71 minutes, in Glarus of almost 1 hour. These findings indicate underserved areas and geographically determined disparities in accessibility of birth settings. The absence of birth centers (that leads to such poor accessibility) might reinforce the negativity that still surrounds birth centers and midwife-led facilities [49], due to an impeded contact between people's life worlds and this particular birth setting.

In fact, research has shown that women's preferences to give birth outside the hospital were often challenged by healthcare professionals, families, and friends and that women needed to actively gather information, if they considered out-of-hospital births [49]. In connection, geographical constraints, opposition in face of a preference for giving birth outside the hospital, and hampered access to information on births in a non-hospital setting represent major barriers to women's reproductive autonomy and frequently urge women to choose a birth hospital

over a birth center. Hence, (still) opting for the latter requires a great extent of confidence to withstand the headwind as well as more resources (e.g. time, financial) to travel to the preferred place of birth and to successfully seek out information. Especially for low-risk nulliparous women who should be advised to give birth in a midwife-led unit such a situation is lamentable [18]. Women should be given a true choice which has to be encouraged by obstetric staff and supported with full, unbiased information [16], even more so during the Covid-19 pandemic when more and more families consider out-of-hospital births. Naturally, women's choices are determined by cultural and social norms, by what is offered to them, and by availability of birth settings, but, as Jane Sandall puts it in an editorial, "availability of maternity services does not guarantee use if women find services hard to access or that services do not meet their needs." [50, p547]. The prevailing situation, however, is often one of limited access to birth centers (as indicated by our analysis), frequently contested preferences to give birth outside the hospital, and a lack of access to adequate information [49].

While accessibility of birth hospitals overall was very good, our analysis identified areas where a notable minority of women of childbearing age faced comparably long travel times. For example, 25% of women of childbearing age in Inner Rhodes had a travel time to the nearest birth hospital of at least 34 minutes. This puts both mother and child at a risk, since it was shown that an increase in travel time to birth hospital is associated with a higher likelihood of early and late neonatal deaths and that residing in a so called maternity care desert (i.e. poor access to birth hospitals and certified midwives) is associated with significantly elevated risks of pregnancy-related and pregnancy-associated mortality [51, 52]. Furthermore, longer travel times to birth hospital might be crucial for home births with complications that require timely transfer to a hospital. In these cases, women might feel uncomfortable with considering a home birth and, as a consequence, are less likely to opt for birthing at home. Against this backdrop, our findings provide valuable insights not only with regard to the supply level of birth centers but also of birth hospitals. However, it has to be noted that for both birth hospitals and birth centers we found a striking variation across cantons which can be attributed to differing topographical and infrastructural actualities, but at the same time is caused by the absence or presence, respectively, of birth hospitals and birth centers.

## Limitations

Our study has several limitations. We did not include neighboring countries' birth facilities. However, it is unlikely that women cross national borders for childbirth. Moreover, the Swiss facility landscape is constantly changing and we cannot preclude that facilities were either closed or opened in the meantime. Nevertheless, the lists of birth hospitals and birth centers were up-to-date (December 2020) and provided by respective authorities. Also, for 429 raster-cells (corresponding to 4.944 women) we were not able to calculate travel times due to topological errors of street network data and incompatibility of different data sources. Anyways, our analysis included 99.8% of Swiss women of childbearing age. Furthermore, the nearest center hypothesis underlying our analysis can be mistaken for individuals who might base the evaluations of their surroundings and preferences regarding birth facilities on different aspects than travel distance. Nonetheless, our analysis offers a detailed spatial insight. Finally, due to unavailable data, we exclusively focused on birth centers and birth hospitals and did not include home births and midwife-led units alongside obstetric units. However, the included birth settings account for the great majority of births in Switzerland [24]. It can be assumed that the proportion of women having a true choice between all four birth settings is even smaller. Analyzing all settings, a UK study found that only 4.2% of women had a true choice [16].

## Conclusion

Our analysis provides a high-resolution insight into the accessibility of birth hospitals and birth centers for Swiss women to an extent that allows reporting travel times at a 100x100 meters level. This spatial resolution surpasses previous studies on accessibility of birth settings. The results are important for evaluating the current status quo and analyzing developments in the future. Since we did not cover home births and midwife-led units alongside an obstetric unit, accessibility of these birth settings should be examined by future research. Also, future research should investigate (factors affecting) Swiss women's attitudes and preferences on place of birth in order to more comprehensively determine the need to reshape the maternal healthcare system (based on both geographical accessibility *and* maternal preferences).

The health system constraints to Swiss women's reproductive autonomy identified by our analysis are likely to even permeate IP-DM. Not being able to choose the preferred birth setting forces a proportion of women to opt for an unwanted birth setting in which conflicts and disagreements with the obstetric staff during childbirth are more likely to arise. Policy-makers, obstetric care providers, and governmental authorities should seek to improve women's choice of place of birth. In their 2021 report, the Aspen Health Strategy Group argues that freestanding birth centers "can reduce maternal mortality and morbidity, providing high-quality, patient-centered, accessible care for the vast majority of pregnancies, which are low-risk" and that they are "a cost-effective alternative to hospital deliveries for low-risk births, with a strong evidence base for better outcomes and higher rates of satisfaction" [37, p12/20]. Moreover, the Covid-19 pandemic and its implications for the health system have been apostrophized as a "'focusing event', or policy window, which may enable midwives and their advocates to shift policy"[6] and as a "catalyst for more integrated maternity care" [53, p1663]. In light of this, and despite the challenges for a nationwide availability of birth hospitals and birth centers, such as a shortage of midwives or increasing national health care costs, our findings provide unique data and strong arguments to (re-)examine the need for further birth centers (and birth hospitals) in specific geographical areas in Switzerland.

## Acknowledgments

The authors acknowledge the support of Prof. Saraswathi Vedam, Birth Place Lab, University of British Columbia, Vancouver, Canada, who contributed to the interpretation of results.

## Author Contributions

**Conceptualization:** Sebastian Rauch, Louisa Arnold, Zelda Stuerner, Michael Rost.

**Data curation:** Sebastian Rauch, Louisa Arnold, Zelda Stuerner, Michael Rost.

**Formal analysis:** Sebastian Rauch, Louisa Arnold, Juergen Rauh, Michael Rost.

**Funding acquisition:** Michael Rost.

**Investigation:** Zelda Stuerner.

**Methodology:** Sebastian Rauch, Juergen Rauh.

**Resources:** Michael Rost.

**Software:** Juergen Rauh.

**Supervision:** Juergen Rauh, Michael Rost.

**Validation:** Zelda Stuerner, Michael Rost.

**Visualization:** Sebastian Rauch, Michael Rost.

**Writing – original draft:** Sebastian Rauch, Michael Rost.

**Writing – review & editing:** Sebastian Rauch, Louisa Arnold, Zelda Stuerner, Juergen Rauh, Michael Rost.

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
