## [Decision Letter · Decision Letter 0]

1 May 2022

PONE-D-21-14065A true choice of place of birth? Swiss women’s access to birth hospitals and birth centers in times of Covid-19PLOS ONE

Dear Dr. Rost,

Thank you for submitting your manuscript to PLOS ONE. After careful consideration, we feel that it has merit but does not fully meet PLOS ONE’s publication criteria as it currently stands. Therefore, we invite you to submit a revised version of the manuscript that addresses the points raised during the review process.

We look forward to receiving your revised manuscript.

Kind regards,

Florian Fischer

Academic Editor

PLOS ONE

Journal Requirements:

4. Please include a caption for figures Map 1-4.

5. Please ensure that you refer to Figures map 1-4 in your text as, if accepted, production will need this reference to link the reader to the figure.

6. We note that Figures Map 1-4 in your submission contain map images which may be copyrighted. All PLOS content is published under the Creative Commons Attribution License (CC BY 4.0), which means that the manuscript, images, and Supporting Information files will be freely available online, and any third party is permitted to access, download, copy, distribute, and use these materials in any way, even commercially, with proper attribution. For these reasons, we cannot publish previously copyrighted maps or satellite images created using proprietary data, such as Google software (Google Maps, Street View, and Earth). For more information, see our copyright guidelines: http://journals.plos.org/plosone/s/licenses-and-copyright.

a. You may seek permission from the original copyright holder of Figure Map 1-4 to publish the content specifically under the CC BY 4.0 license.  

Reviewers' comments:

Reviewer's Responses to Questions

**Comments to the Author**

1. Is the manuscript technically sound, and do the data support the conclusions?

Reviewer #1: Yes

Reviewer #2: Yes

2. Has the statistical analysis been performed appropriately and rigorously? 

Reviewer #1: Yes

Reviewer #2: Yes

3. Have the authors made all data underlying the findings in their manuscript fully available?

Reviewer #1: No

Reviewer #2: No

4. Is the manuscript presented in an intelligible fashion and written in standard English?

Reviewer #1: Yes

Reviewer #2: Yes

5. Review Comments to the Author

Reviewer #1: Thanks for allowing me to review your interesting manuscript. The results on accessibility are very interesting and I suspect could be replicated in many other countries. Given the relatively larger number of birth hospitals when compared to birth centres the difference in travel times is not that surprising.

However, choice of place of birth is dependent on many more things other than accessibility based on travelling times. For example, many birth centres (in my own experience) are limited in capacity and will only accept a limited number of bookings each month. Sometimes this is based on the models of care and the number of women each midwife or group of midwives can care for. Do you have any data on capacity of each birth centre and to what extent this may also limit choices?

Reviewer #2: The topic of the article is very interesting, since it is ne topic to be discussed. I support to accept this article because it offers new perspective and novelty of the topic. I would recommend to accept to publish this article.

6. PLOS authors have the option to publish the peer review history of their article (what does this mean?). If published, this will include your full peer review and any attached files.

Reviewer #1: **Yes: **David Ellwood

Reviewer #2: No

---

## [Author Response · Author response to Decision Letter 0]

17 May 2022

See attached document

(Please note that maps 1-4 were created by the authors themselves (and not obtained from a third-party) and thus are not copyrighted. Therefore, these maps can be published in their current form without any permission or consent from a possible copyright holder (see first comment in responses to reviewers).)

---

## [Decision Letter · Decision Letter 1]

20 Jun 2022

A true choice of place of birth? Swiss women’s access to birth hospitals and birth centers in times of Covid-19

PONE-D-21-14065R1

Dear Dr. Rost,

We’re pleased to inform you that your manuscript has been judged scientifically suitable for publication and will be formally accepted for publication once it meets all outstanding technical requirements.

Kind regards,

Florian Fischer

Academic Editor

PLOS ONE

Additional Editor Comments (optional):

Reviewers' comments:

Reviewer's Responses to Questions

**Comments to the Author**

1. If the authors have adequately addressed your comments raised in a previous round of review and you feel that this manuscript is now acceptable for publication, you may indicate that here to bypass the “Comments to the Author” section, enter your conflict of interest statement in the “Confidential to Editor” section, and submit your "Accept" recommendation.

Reviewer #1: All comments have been addressed

2. Is the manuscript technically sound, and do the data support the conclusions?

Reviewer #1: Yes

3. Has the statistical analysis been performed appropriately and rigorously? 

Reviewer #1: Yes

4. Have the authors made all data underlying the findings in their manuscript fully available?

Reviewer #1: Yes

5. Is the manuscript presented in an intelligible fashion and written in standard English?

Reviewer #1: Yes

6. Review Comments to the Author

Reviewer #1: Thank you for addressing all of my previous comments after the initial review. I have no further comments or questions.

7. PLOS authors have the option to publish the peer review history of their article (what does this mean?). If published, this will include your full peer review and any attached files.

Reviewer #1: **Yes: **Professor David Ellwood

---

## [Editor Report · Acceptance letter]

24 Jun 2022

PONE-D-21-14065R1 

A true choice of place of birth? Swiss women’s access to birth hospitals and birth centers 

Dear Dr. Rost:

I'm pleased to inform you that your manuscript has been deemed suitable for publication in PLOS ONE. Congratulations! Your manuscript is now with our production department. 

Kind regards, 

on behalf of

Dr. Florian Fischer 

Academic Editor

PLOS ONE